# Sub-seasonal and Spatial Variations in Ozone Formation and Cocontrol Potential for Secondary Aerosols in the Guanzhong Basin, Central China

Ruonan Wang<sup>1,2</sup>, Ningning Zhang<sup>1\*</sup>, Jiarui Wu<sup>1,2</sup>, Qian Jiang<sup>1</sup>, Jiaoyang Yu<sup>1,2</sup>, Yuxuan Lu<sup>1</sup>, and Xuexi Tie<sup>1</sup>

<sup>1</sup>State Key Laboratory of Loess Science, Institute of Earth Environment, Chinese Academy of Sciences, Xi'an 710061, China <sup>2</sup>National Observation and Research Station of Regional Ecological Environment Change and Comprehensive Management in the Guanzhong Plain, Xi'an 710049, China.

Correspondence to: Ningning Zhang (zhangnn@ieecas.cn)

Abstract: Tropospheric ozone (O<sub>3</sub>) pollution in warm seasons has become the key air-quality issue in the Guanzhong Basin (GZB), threatening human health despite prior successes in particulate matter control. Understanding how O<sub>3</sub> formation regimes (OFR) and secondary aerosol (SA) formation regimes (SAFR) evolve with time and space is critical for designing coordinated control strategies. Long-term near-surface observations (2014-2024) are combined with high-resolution WRF-Chem simulations for May-August 2022, employing scenario-based EKMA curves and source-apportionment diagnostics to resolve sectoral contributions. Results indicate a sub-seasonal OFR progression from VOCs-limited in early summer to transitional in midsummer and NO<sub>x</sub>-limited in late summer, with anthropogenic contribution to the maximum daily averaged 8-h (MDA8) O<sub>3</sub> increasing from 32.8% in May to 55.2% in July and biogenic share peaking 18.7% in July. SAFR follows a distinct cycle with NO<sub>x</sub>-limited in May, VOCs-limited in June, and transitional behavior thereafter. Traffic and industrial emissions are the dominate anthropogenic divers for both O<sub>3</sub> and SA. These patterns highlight phases of synergistic control, where anthropogenic VOCs mitigation in June and NO<sub>x</sub> mitigation in August maximize co-benefits while minimizing tradeoffs. This study integrated dynamic OFR/SAFR diagnostics with sectoral emission inventories can provide insights into pathways toward seasonally adaptive, city-specific air quality management in the GZB.

### 1 Introduction

Tropospheric ozone (O<sub>3</sub>), despite constituting only approximately 10% of total atmospheric O<sub>3</sub>, poses a significant threat to human health and ecosystem integrity (Valacchi et al., 2015; Feng et al., 2022). Long-term observations at regional background stations reveals a persistent upward trend in near-surface O<sub>3</sub> concentrations in China in recent decades (Wang et al., 2009; Sun et al., 2016; Xu et al., 2016). The nationwide implementation of the "Air Pollution Prevention and Control Action Plan" since September 2013 has achieved notable success in mitigating fine particulate matter (PM<sub>2.5</sub>) pollution (Zheng et al., 2018; Zhang et al., 2019). However, O<sub>3</sub> pollution has emerged as a paramount challenge for air quality management in China during the warm season (May-August), driving extensive research on formation mechanisms and PM<sub>2.5</sub> co-control strategies (Li et al., 2019; Li et al., 2020; Liu et al., 2020; Wang et al., 2022; Wang et al., 2023; Wang et al., 2024).

Near-surface O<sub>3</sub> is a secondary pollutant formed through intricate photochemical reactions involving nitrogen oxides (NO<sub>X</sub>) and volatile organic compounds (VOCs) under sunlight (Chapman, 1930; Chameides et al., 1992; Wang et al., 2017). The complex photochemical pathways of O<sub>3</sub> formation pose a significant barrier to effective O<sub>3</sub> pollution control, resulting in persistent high O<sub>3</sub> concentrations. The inherent nonlinearity in tropospheric O<sub>3</sub> formation necessitates the assessment of its Formation sensitivity Regime (OFR), which acts as a prerequisite for implementing effective control strategies. OFR is categorized into NO<sub>X</sub>-limited, VOCs-limited, and transitional regimes according to the relationship of O<sub>3</sub> with its precursors (Lu et al., 2019). In NO<sub>X</sub>-limited regimes, O<sub>3</sub> production is primarily terminated by self-reactions of peroxyl radicals, reducing NO<sub>X</sub> emissions effectively lowers O<sub>3</sub>. Conversely, in VOCs-limited regimes, termination occurs mainly via NO<sub>2</sub> + HO· reactions; reducing NO<sub>X</sub> can slow HO· loss and reduce O<sub>3</sub> titration, resultantly increasing O<sub>3</sub> levels (Jenkin and Clemitshaw, 2000). Transitional regimes exhibit comparable sensitivity to both precursors. Current OFR assessment methods include indicator approaches (Sillman et al., 1995; Martin et al., 2004; Li et al., 2011), observation-based models (OBM) (Cardelino and Chameides, 1995; Wang et al., 2017; Song et al., 2022), and emission-based models (EBMs) utilizing three-dimensional chemical transport models (CTMs) (Li et al., 2018; Xu et al., 2022). Despite computational demands, EBMs provide a robust framework for resolving OFR across horizontal and vertical dimensions and over time, enabling direct investigation of OFR under varying meteorology and emission reduction scenarios (Ou et al., 2016; Wang et al., 2021).

Owing to the common precursors and complex interplay, synergistic control of PM<sub>2.5</sub> and O<sub>3</sub> has emerged as an essential priority for China's air pollution mitigation in recent years (Xiao et al., 2021; Du et al., 2024; Kong et al., 2024). In addition to the common share of precursors (NO<sub>X</sub> and VOCs), the ambient O<sub>3</sub> and its photochemical derivative, hydroxyl radicals (·OH), constitute the major oxidants that oxidize precursors to form secondary aerosols (SA), such as sulfates, nitrates, and secondary organic aerosols (SOA). Moreover, hydroperoxyl radicals (HO<sub>2</sub>·), as a critical participate in O<sub>3</sub> production, whose concentration can be influenced by heterogeneous uptake on wet aerosol surfaces (Li et al., 2019; Wang et al., 2022). Furthermore, aerosol-radiation and aerosol-cloud interactions alter solar radiation and temperature in the atmosphere, thereby

influencing O<sub>3</sub> photochemical production (Baró et al., 2017; Li et al., 2019; Wu et al., 2020a). Therefore, optimizing precursor emission reductions requires not only understanding OFR but also the SA formation sensitivity regime (SAFR) to NO<sub>X</sub> and VOCs. Integrating OFR and SAFR analyses under diverse meteorological conditions is critical to identifying co-beneficial pathways for simultaneously improving air quality.

The Guanzhong Basin (GZB), a key air pollution control area in central China, experiences severe warm-season O<sub>3</sub> pollution. Bei et al. (2022) have revealed that increased summertime unfavorable synoptic conditions are responsible for the deteriorated O<sub>3</sub> pollution in the GZB from 2014 to 2018. Biogenic emissions from extensive forests in Qinling mountains, situated south of the GZB, could interact with abundant anthropogenic emissions within the GZB when southerly winds prevail, facilitating O<sub>3</sub> formation in the region (Feng et al., 2016; Li et al., 2018; Dai et al., 2024). However, studies remain limited on the O<sub>3</sub> formation characteristics and the synergistic pollutants abetment in the region (Wang et al., 2022). This study employs a high-resolution online-coupled model to comprehensively resolve spatiotemporal patterns in warm-season O<sub>3</sub> formation characteristics and provide insights into O<sub>3</sub> pollution mitigation and synergistic air quality management in the GZB.

# 2 Methodology

## 2.1 The WRF-Chem model

The WRF-Chem model (Weather Research and Forecasting model with Chemistry) (Grell et al., 2005) utilized in this study is a specific version modified by Li et al (2010; 2011a; 2011b; 2012) and has been extensively used for regional simulations of air pollutants in the GZB. Simulations focus on the atmospheric processes over the GZB in the warm season of 2022. This period is selected for its exceptionally high O<sub>3</sub> pollution (with maximum daily averaged 8-h (MDA8) O<sub>3</sub> concentration of 134 µg m<sup>-3</sup>, +7% above the 2020-2024 mean), enabling a detailed analysis of photochemical mechanisms and co-control pathways. Figure 1 shows the model simulation domain encompassing the GZB and its surrounding regions, with particular focus on five major cities: Xi'an (XA, provincial capital), Xianyang (XY, undergoing rapid industrialization and urbanization), Weinan (WN, energy hub), Tongchuan (TC), and Baoji (BJ). Detailed model configuration can be found in S1 and Table S1 of the Supplementary Information (SI). This setup constitutes the base simulation, which serves as the reference for subsequent sensitivity scenarios.

# 2.2 Observations and Statistics for Comparisons

Observations of criteria pollutants (PM<sub>2.5</sub>, NO<sub>2</sub>, O<sub>3</sub>, SO<sub>2</sub> and CO) released by China's Ministry of Ecology and Environment (MEE) are used for characterizing recent warm-season (2014-2024) air quality trends across the GZB and validating the simulated air pollutants. Meteorological parameters, including 2 m temperature, relative humidity, and 10-m wind speed/direction, measured at Jinghe station in the GZB (34.26°N and 108.58°E) are employed to evaluate the model performance in simulating synoptic conditions. Model performance of the base simulation is evaluated against observations

using statistical metrics including the mean bias (MB), root mean square error (RMSE), and the index of agreement (IOA, shown in S2 of the SI).

#### 2.3 Sensitivity simulations

Building upon the base simulation, an emission reduction matrix comprising 121 scenarios is then designed to develop EKMA diagrams to determine the O<sub>3</sub> and SA formation regime (Figure S1). These scenarios include reductions of NO<sub>X</sub> and AVOCs emissions from 0% to 100% with an interval of 10%. Given that the biogenic VOCs (BVOCs) emissions are uncontrollable, only the AVOCs emission reduction is taken into consideration. It does not mean the insignificance of BVOCs in the O<sub>3</sub> formation. However, changes of plant cover and emissions as well as the O<sub>3</sub> uptake capacity of plants all affect the near-surface O<sub>3</sub> concentration through ecosystem-atmosphere interactions (Lin et al., 2020). Furthermore, the brute force method (BFM) is used to evaluate the O<sub>3</sub> and SA contributions of industry, power, residential, transportation and biogenic sources in the GZB and five cities during the warm season of 2022. The BFM directly closes or cuts out one emission source in the base simulation to calculate its contribution to the O<sub>3</sub> and SA formation (Dunker et al., 1996).

# 3 Results and Discussion

# 3.1 Air quality during warm seasons in the GZB

To understand the evolving characteristics of summertime photochemical pollution in the GZB, we first examine recent trends in near-surface O<sub>3</sub> and PM<sub>2.5</sub> concentrations during the warm seasons from 2022 to 2024. This period represents the most recent stage of air quality management in the region and provides insight into ongoing challenges posed by secondary pollutants. Under China's sequential air pollution policies—the Air Pollution Prevention and Control Action Plan (2013-2017; State Council, 2013), the Three-Year Action Plan to Win the Blue-Sky War (2018-2020; State Council, 2018), and the Air Quality Continuous Improvement Action Plan (2023-2025; MEE, 2023)—the GZB has achieved sustained PM<sub>2.5</sub> reductions. Warm-season PM<sub>2.5</sub> concentrations have decreased from approximately 43.1 μg m<sup>-3</sup> in 2014 to around 21.6 μg m<sup>-3</sup> in 2024, representing a nearly 50% reduction (Fig. 2a). In contrast, the mean MDA8 O<sub>3</sub> concentration during the warm season has increased from 96.0 μg m<sup>-3</sup> in 2014 to roughly 132.7 μg m<sup>-3</sup> in 2024, with a substantial increase from 2014 to 2017, a decrease from 2017 to 2020, followed by a renewed upward trend thereafter (Fig. 2b). While the substantial alleviation in the particulate pollution is mainly due to the anthropogenic emission mitigation, O<sub>3</sub> pollution has worsened in the region primarily driven by increased unfavorable conditions and secondarily by changes in anthropogenic emissions (Bei et al., 2022). Near-surface observations have revealed persistent O<sub>3</sub> pollution during the warm season, with frequent exceedances of the national ambient air quality standard for MDA8 O<sub>3</sub> (160 μg m<sup>-3</sup>, as shown in Table 1). Year-to-year fluctuations in mean MDA8 O<sub>3</sub> levels are governed by the frequency of exceedances, which rise from 8.3% in 2014 to 36.7% in 2017, fall to 13.6% in 2020, and climb

again to 27.0% in 2024; the magnitude of those exceedances remains relatively stable, ranging from 178 to 189  $\mu$ g m<sup>-3</sup> (Table 1).

The monthly evolution of O<sub>3</sub> concentrations reveals a pronounced sub-seasonal pattern in the GZB. June consistently emerges as the peak month for both MDA8 O<sub>3</sub> levels and exceedance frequency (Fig. S5 and Table S2). During the warm seasons from 2014 to 2024, the mean MDA8 O<sub>3</sub> concentration in June reaches approximately 132 µg m<sup>-3</sup>, with over 30% of days with O<sub>3</sub> exceedances. The peak corresponds closely with meteorological conditions typically observed in early summer, when the region experiences strong solar radiation, elevated temperatures, and relatively low precipitation conditions that are highly conducive to photochemical O<sub>3</sub> formation (Pay et al., 2019). The early-summer O<sub>3</sub> peak observed in the GZB is consistent with reports from northern China, where June maxima are evident in the Beijing–Tianjin–Hebei (BTH) region and June–July peaks prevail across the North China Plain (Han et al., 2020; Li et al., 2020). In contrast, in southern China, O<sub>3</sub> subseasonality is strongly modulated by subtropical high-pressure systems, the East Asian monsoon, typhoon passages, and land–sea breeze circulations. As a result, the peak O<sub>3</sub> episodes are tended to emerge in May or during the late summer to early autumn (September–October) in the Yangtze River Delta (YRD) and Pearl River Delta (PRD) (Han et al., 2020; Xu et al., 2020; Ouyang et al., 2022). These regional contrasts highlight that while the precise timing of seasonal O<sub>3</sub> peaks is shaped by local climate and meteorology, the emergence of a distinct sub-seasonal maximum is a robust feature across China's major urban clusters. This underscores the sensitivity of regional air quality to meteorological transitions and highlights the need for sub-seasonally adaptive control strategies, in particularly during the high-risk period.

How these basin-wide trends manifest at the city scale is then explored. O<sub>3</sub> variations in all five cities mirror the overall pattern in GZB, with MDA8 O<sub>3</sub> concentrations rising from 2014 to a maximum in 2017 and then dipping through 2020 before climbing again to 2024 levels. Cities of XA, XY, and WN, located in the central GZB, have the relatively high O<sub>3</sub> levels, with the mean MDA8 O<sub>3</sub> concentration ranging of 128-129 µg m<sup>-3</sup> during the warm seasons of 2014-2024. XY has experienced the fastest MDA8 O<sub>3</sub> concentration increase of 6.3 µg m<sup>-3</sup> yr<sup>-1</sup>, nearly twice the basin average of 3.7 µg m<sup>-3</sup> yr<sup>-1</sup>, which likely reflects rapid urban expansion and increasing local precursor emissions. The city also records the highest fraction of exceedance days, averaging 28.7% of warm-season days during 2014–2024 and peaking at 49.2% in 2017 (Table 1). XA is also characterized by the sever O<sub>3</sub> pollution comparing with other cities, with a growth rate of 3.8 µg m<sup>-3</sup> yr<sup>-1</sup> and exceedances on 25.9% of the warm-season days, consistent with its role as the region's primary emission hub. By contrast, BJ, situated on the western edge and often upwind of the basin core, had the lowest O<sub>3</sub> burden, with the mean MDA8 O<sub>3</sub> concentration of 113.3 µg m<sup>-3</sup> and only 10.9% exceedances of warm-season days during 2014-2024. These spatial differences underscore the importance of both local precursor controls in rapidly urbanizing cities and regional transport pathways in shaping O<sub>3</sub> pollution across the GZB.

Generally, the long-term observations reveal not only a basin-wide deterioration in O<sub>3</sub> pollution but also strong subseasonal and spatial heterogeneity, highlighting the necessity of process-based modeling to resolve spatiotemporal patterns in warm-season O<sub>3</sub> formation characteristics and to provide insights into air pollution mitigation in the GZB.

### 3.2 Model validation

The meteorological simulations during the warm season of 2022 demonstrate excellent reproduction of diurnal temperature patterns (IOA = 0.99) despite slight overestimation biases (+0.4 °C), while relative humidity variations are well captured (IOA = 0.95) with marginal underestimation (-1.8%, Fig. S2). Spatially, the model reproduces key air pollutants distributions across the GZB against measurements (Fig. S3). Elevated PM<sub>2.5</sub> and O<sub>3</sub> in the eastern and central regions arise from northeasterly transport, weak winds over the central basin plain that favor accumulation of air pollutants, and southerly flows carrying BVOCs from the Qinling forests. Temporally, simulated air pollutant concentrations show good agreement with observations with IOAs all exceeding 0.5 (Fig. S4). The model's good performance in replicating synoptic-scale meteorological processes and associated air pollutants warrants its suitability for mechanistic analysis. Comprehensive validation of the WRF-Chem model performance is detailed in S3 of the SI.

## 3.3 Spatiotemporal Patterns of O<sub>3</sub> Sensitivity from EKMA Analysis

O<sub>3</sub> formation in the planetary boundary layer (PBL) is a complex and nonlinear process driven by sunlight acting on NO<sub>X</sub> and VOCs precursors. Figures 3 and 4 present EKMA diagrams for four high-O<sub>3</sub> pollution episodes from May to August 2022 in urban areas of the GZB and its five cities, respectively. These diagrams depict O<sub>3</sub> isopleths for OFR identification, derived from sensitivity simulations with systematically reduced NO<sub>X</sub> and AVOCs emissions. The ridge line (red lines) delineates the boundary between these regimes: scenarios above it lie in the VOCs-limited regime (O<sub>3</sub> falls more with AVOCs cut), those below in the NO<sub>X</sub>-limited regimes, and scenarios near the line are transitional regimes (mixed sensitivity). The upper-right corner (100% AVOCs, 100% NO<sub>X</sub> emissions) represents the current emission scenario, whose location relative to the ridge line determines the prevailing sensitivity regime.

EKMA curves reveal pronounced spatiotemporal shifts in OFR. In May and June, the GZB as a whole was predominantly VOCs-limited, indicating that AVOCs reductions would substantially lower O<sub>3</sub> concentrations, whereas moderate NO<sub>x</sub> cuts could exacerbate O<sub>3</sub> pollution (Figs. 3a and 3b). City by city, however, sensitivity varied. O<sub>3</sub> formation in BJ lies in a VOCs-sensitive regime close to the transitional zone, where initial reductions in either NO<sub>x</sub> or AVOCs emissions lead to decreases in MDA8 O<sub>3</sub> concentrations (Fig. 4a5). O<sub>3</sub> concentrations exhibit greater sensitivity to reductions in AVOCs emissions before precursor emissions being cut by approximately 50%. WN occupies the most NO<sub>x</sub>-saturated zone of the GZB in May and June under current emissions: with 100% AVOCs emissions, a 60% cut in NO<sub>x</sub> emissions results in 11.7% and 9.3% increases in urban MDA8 O<sub>3</sub> concentrations in May and June, respectively (Figs. 4a3 and 4b3).

By July, OFR in urban areas of the GZB shifts toward the transitional regime (Fig. 3c). Under current AVOCs emissions, a NO<sub>X</sub> reduction exceeding 10% is sufficient to achieve a notable decrease in MDA8 O<sub>3</sub> concentration. All five cities show movement toward less VOCs-sensitive regimes (Fig. 4c). O<sub>3</sub> formation in XA and WN remains primarily AVOCs-controlled, whereas it in XY, TC, and BJ becomes transitional. Ordering the cities with OFR varying from most VOCs-sensitive to most NO<sub>X</sub>-sensitive conditions yields the sequence that WN > XA > TC > XY > BJ. In August, OFR in the GZB enters a NO<sub>X</sub>-limited regime (Fig. 3d): a 40% NO<sub>X</sub> emission mitigation delivers an average MDA8 O<sub>3</sub> decrease of 11.1%, whereas an equal AVOCs reduction yields only a 3% decrease. OFRs in all cities except in WN (transitional) are NO<sub>X</sub>-limited (Fig. 4d), indicating that initial NO<sub>X</sub> reductions represent the most effective mitigation pathway to alleviate O<sub>3</sub> pollution.

The temporal OFR shifts can be attributed to concurrent changes in chemistry and meteorology. The AVOCs/NOx emission ratios are relatively stable (0.27-0.34; Table S3) during the warm season. This variation cannot explain the stronger NOx sensitivity detected in July-August, indicating that anthropogenic precursor ratios alone do not fully account for the seasonal OFR shift. The most important change from May to August is the intensification of solar radiation and the resultant increase in air temperature. Firstly, BVOCs emissions are dependent on solar radiation and air temperature, so increased solar radiation and air temperature in mid-late summer boost BVOCs emissions, providing more background VOCs and pushing the O<sub>3</sub> formation toward NOx-sensitive. Secondly, enhancement of solar radiation and higher temperature accelerate photochemical reactions. In addition, higher temperatures favor a deeper PBL, which enhances vertical mixing and can entrain O3-rich air from aloft while diluting near-surface precursor concentrations and thus altering local precursor ratios. Near-surface O3 tends to increase as the PBL deepens from shallow to moderate heights, peaking at intermediate mixing depths of approximately 900-1800 m (Wang et al., 2023). In urban areas of the GZB, the mean PBL height (PBLH) during 11:00-18:00 BJT rises from 1382 m in May to 1720 m in June, then falls to 1412 m in July and 1406 m in August, consistent with the maximum MDA8 O3 level in June. Simulations indicate that HOX radical concentrations increase while near-surface NOX levels decrease from May to August in urban areas of the GZB (Fig. 5 and Table S4). These changes are closely linked to enhanced BVOCs emissions, intensified atmospheric photochemistry and PBL development, which alter relative balance of the O<sub>3</sub> precursor levels. Consequently, HO<sub>x</sub>-loss becomes increasingly dominated by self-reaction of peroxyl radicals rather than HO· + NO<sub>2</sub> termination, further shifting O<sub>3</sub> production to be more NO<sub>X</sub>-sensitive. The similar transition trend has been found in previous studies. Wu and Xie (2017) have discussed occurrence of a switch from a NOx-saturated to NOx-sensitive O<sub>3</sub> formation regime in most suburban and rural areas in China when summer arrives. Ou et al. (2016) have proposed that O<sub>3</sub> formation shifts toward VOCs-limited conditions in the PRD from summer to autumn. Sun et al. (2018) have used highresolution observations in eastern China to show that the photochemical regime during spring and summer tends toward NOxlimited or mixed sensitivity, while in autumn and winter it shifts toward VOCs-limited conditions. Our study extends this understanding by resolving OFR transitions at sub-seasonal (monthly) and city-specific levels, offering feasible insights for dynamic emission control.

Spatially, OFRs varied markedly among cities in relation to local VOCs (AVOCs + BVOCs) / NOx emission ratios (Fig. 6). In early summer, XA, XY, WN and TC all have relatively lower ratios (within 2 to 7) and exhibit VOCs-limited regimes, indicating their high NOx emissions make O3 formation strongly VOCs-constrained. By contrast, BJ generally falls within the NOx-limited or transitional regimes (except in June) with relatively high VOCs/NOx ratio ranging from 11 to 29, suggesting comparatively lower NO<sub>X</sub> concentration or/and relatively higher VOCs levels. OFR in WN exhibits the most NO<sub>X</sub>saturated (VOCs-limited) characteristics, even though its VOCs/NOx emission ratio is not the lowest—in part because WN lies on the eastern margin of the GZB and is susceptible to pollutant transport from heavily polluted regions in Henan and Shanxi (Li et al., 2021). This inflow elevates local NO<sub>X</sub> (with relatively longer atmospheric lifetime than that of reactive VOCs) concentrations relative to VOCs, emphasizing the necessity of targeted AVOCs reductions before aggressive NOx cuts can effectively mitigate O<sub>3</sub> pollution in this city. These intra-region contrasts underscore that control strategies must be tailored to local chemistry. Spatially, OFRs varied markedly among cities in relation to local VOCs/NOx emission ratios (Fig. 6). In early summer, XA, XY, WN, and TC all had low VOCs/NO<sub>x</sub> ratios (~2-7) and exhibited VOCs-limited regimes, indicating their high NO<sub>x</sub> emissions made ozone formation strongly VOCs-constrained. By contrast, BJ's much higher VOCs/NO<sub>x</sub> ratio (11-29) placed it in NO<sub>x</sub>-limited or transitional regimes. Notably, WN was the most NO<sub>x</sub>-saturated city, even though its VOCs/NO<sub>x</sub> was not the absolute lowest – in part because WN lies on the GZB's eastern margin and is affected by pollutant transport from heavily polluted northern regions. The inflow of NOx-rich air elevates local NOx relative to VOCs, reinforcing VOCs-limited chemistry in WN. These intra-region contrasts underscore that control strategies must be tailored to local chemistry. Similar patterns occur elsewhere: Ren et al. (2022) found heavily polluted Chinese cities (e.g. Wuhan, Xi'an) were strongly VOCslimited, whereas suburb and rural areas were NOx-limited. Likewise, Yu et al. (2025) diagnosed Zhengzhou's ozone regime as primarily VOCs-limited with an optimal VOCs: NOx reduction ratio of ~2.9:1, echoing the high VOCs-sensitivity we see in industrial cities.

Overall, these spatial and sub-seasonal OFR shifts highlight the necessity of dynamic, month-specific O<sub>3</sub> control strategies in the GZB. The pronounced VOCs-limited conditions in early summer call for prioritizing AVOCs control, especially in cities with sever O<sub>3</sub> pollution like WN and XA, while the transition to NO<sub>X</sub>-limited conditions by late summer favors NO<sub>X</sub>-focused measures. Such temporally and spatially optimized approaches could enhance the efficacy of regional O<sub>3</sub> mitigation and help avoid unintended increases during seasonal transitions.

# 3.4 Sectoral Contributions to Warm-Season O<sub>3</sub> Production

Quantifying sectoral drivers of warm-season O<sub>3</sub> production bridges the sensitivity diagnostics from Section 3.2 with operational emission control design. The MDA8 O<sub>3</sub> contributions of industrial, residential, traffic, power plants and biogenic sources are assessed from May to August using the BFM (Figs. 7 and 8).

The attribution results show pronounced spatiotemporal heterogeneous across the GZB. At the basin scale, the contribution of all anthropogenic sources increases from 32.8% in May to 55.2% in July, then declines to 48.0% in August (Fig. 7). The rise is largely driven by increasing industrial and traffic influence: industrial emissions contribute 5.4% to MDA8 O<sub>3</sub> in May, rising to 11.8% in July, while traffic contributions increase from 5.8% to 17.0% during the same period (Fig. 8 and Table 3). Because NO<sub>X</sub> and AVOCs emissions peak in June and then decline, the continued anthropogenic contribution growth through July indicates that enhanced photochemical activity under strong solar radiation and rising BVOCs emissions further amplify O<sub>3</sub> formation. Biogenic contributions notably rise from 9.7% in May to 18.7% in July then slightly decline to 16.7% in August (Fig. 7 and Table 3), consistent with elevated BVOCs emissions under warmer, sunnier conditions. These subseasonal dynamics are consistent with findings from other Chinese urban clusters, such as the BTH, YRD and PRD, where rising BVOCs emissions in summer have been shown to enhance photochemical reactivity and partially offset gains from anthropogenic VOCs and NO<sub>X</sub> reductions, thereby promoting shifts of OFR toward NO<sub>X</sub>-limited or transitional regimes (Wu et al., 2020; Zhao et al., 2022; Gao et al., 2025; Wang et al., 2025).

Spatial contrasts across the five cities further illustrate how emission profiles interact with chemical regimes and guide targeted mitigation polices. XA and XY show the largest anthropogenic O3 shares, rising from 37.1% in May to 58.7% in July in XA, and from 35.7% to 59.2% in XY (Fig. 7 and Table 3). In these urban cores, industrial and traffic emissions are dominant, contributing approximately 9-12% (May) and 26-31% (June-August) to MDA8 O3 concentrations (Fig. 8). TC and BJ show intermediate anthropogenic contributions. Traffic dominates their O<sub>3</sub> production (7.1-15.9% in TC and 7.9-18.3% in BJ), and industrial emissions are the secondary contributor to the O3 levels. Given their VOCs-limited regime (Section 3.2) and higher industrial AVOCs emissions (Table S3) in May-June, prioritizing reductions in industrial AVOCs is advisable to prevent O3 rebound. As the OFR shifts toward transition and NOx-limited conditions in late summer, mitigation efforts should be combined with traffic emissions due to the higher NOx share from vehicular exhaust. In industrial regions, for instance, Dai et al. (2025) have showed that oxygenated VOCs (OVOCs) contribute a high proportion (~30-37.8%) of VOCs pools in industrial cities, enhancing radical production and O3 sensitivity; thus, control strategies must consider VOCs speciation, not just total amounts. By contrast, WN exhibits much lower anthropogenic influence (13.9-34.7%, Fig. 7 and Table 3), with power plant emissions exerting a net consuming effect on local O<sub>3</sub>. This is attributed to strong local NO titration and enhanced termination pathways reduce OH/HO2 recycling. Similar suppression effects have been reported in Zhao et al. (2025) that power plant NO<sub>X</sub> emissions contribute negative O<sub>3</sub> signals in certain regions, indicative of O<sub>3</sub> titration under saturating NO<sub>X</sub>. Meanwhile, analyses in the YRD have showed that during emission reductions, weakened NO titration can drive O3 increases (~20% of the rise attributed to reduced NO titration) (Wang et al., 2022). Mechanistic modeling also indicates that NOx reductions in VOCs-limited areas may lead to O<sub>3</sub> increases, unless VOCs controls are pursued simultaneously (Dai et al., 2024; Tang et al., 2021). Therefore, aggressive NO<sub>X</sub> cuts alone risk raising O<sub>3</sub> unless paired with targeted AVOCs reductions in WN.

These variations highlight that a uniform, time-invariant control strategy is insufficient. Instead, effective O<sub>3</sub> mitigation in the GZB demands sub-seasonally adaptive, sector-specific emission controls. In early summer, dominant VOCs sensitivity suggests focusing on industrial AVOCs; in mid to late summer, integrating AVOCs and traffic NO<sub>X</sub> controls aligns better with evolving regimes; and in regions like WN, VOCs-targeted strategies must accompany any NO<sub>X</sub> reductions. Recent studies similarly argue that tailoring precursor reductions to local O<sub>3</sub> regimes yields greater benefits than uniform cuts. For example, Zhu et al. (2022) showed that in the NCP and YRD, formation regimes have shifted from VOCs-limited toward transitional or NO<sub>X</sub>-limited states as atmospheric oxidation increases. Wang et al. (2025) reported pronounced vertical and spatial differences in OFRs in eastern Chinese cities, implying that a one-size-fits-all control is often suboptimal. Li et al. (2024) have used FNR (HCHO/NO<sub>2</sub>) diagnostics at multiple altitudes to reveal that optimal precursor strategies vary by vertical layer in the BTH.

## 3.5 Synergistic Control of O<sub>3</sub> and Secondary Aerosols

While summertime air quality in the GZB is dominated by O<sub>3</sub> pollution, SA remain non-negligible even when PM<sub>2.5</sub> is relatively low. The interaction between O<sub>3</sub> and SA is multifaceted: O<sub>3</sub> and SA are closely linked through shared precursors and atmospheric oxidation processes. Elevated O<sub>3</sub> levels enhance the formation of SA by increasing the oxidizing capacity of the atmosphere, thereby accelerating the conversion of gaseous precursors into both inorganic and organic aerosol components. Yu et al. (2025) have noted that secondary organic carbon increases when O<sub>3</sub> exceeding 50 μg m<sup>-3</sup>, indicating stronger photochemistry yields more SOA. Conversely, by scattering or absorbing radiation, aerosols can change the intensity and spectral distribution of light, thereby modulating photolysis rates (Wu et al., 2019; Wu et al., 2020). In addition, aerosol surfaces can facilitate or suppress specific chemical reactions—such as the conversion of NO<sub>2</sub> to nitrate or the uptake of hydroperoxyl radicals—which in turn influence O<sub>3</sub> production and loss pathways (Li et al., 2019; Wang et al., 2022). These bidirectional interactions mean that changes in one pollutant often propagate to the other, making it essential to consider them within a unified management framework. During the warm season, when photochemical activity peaks and O<sub>3</sub> dominates air pollution, incorporating SA considerations into O<sub>3</sub>-focused control strategies can provide additional air quality benefits and help prevent counterproductive effects on pollutant levels.

Warm-season SAFR diagnostics show clear seasonal shifts (Figs. 3 and 9). At the basin scale, SA is NO<sub>X</sub>-limited in May, with reductions in NO<sub>X</sub> emissions delivering roughly three times the concentration decreases in SA compared to equivalent mitigation in AVOCs emissions (Fig. 3e). In June, the regime shifts to VOCs-limited, with benefits from AVOCs emission reductions outweighing those from NO<sub>X</sub> controls (Fig. 3f). In July, SAFR exhibits a transitional state with sensitivity varying by location, whereas in August it reverts to a transitional regime tilted toward NO<sub>X</sub> sensitivity (Figs. 3g and 3h). City-level patterns follow this broad seasonal evolution but reveal important local deviations (Fig. 9): in June, XA, XY, WN, and TC are VOCs-limited, whereas BJ remains transitional. SAFR of WN stands out by retaining VOCs sensitivity into July, albeit weaker than in June, while other cities transition earlier. By August, SAFR of all cities exhibit transitional responses with a leaning

toward NO<sub>X</sub> sensitivity, indicating that late summer presents an opportunity for NO<sub>X</sub>-focused co-control. Source attributions show anthropogenic sources dominate SA formation (63-87%) across the GZB, with traffic and industry are principal contributions (Table 4). Power plant emissions are particularly influential in WN, where their NO<sub>X</sub> emissions drive elevated SA formation, while simultaneously exerting a local O<sub>3</sub>-suppressing effect. Residential emissions contribute moderately, ranging from 7–22% across cities and months, while biogenic sources play a relatively minor role in SA formation.

These findings point to several opportunities for achieving synergistic O<sub>3</sub>–SA control. In May, SAFR is strongly NO<sub>x</sub>-limited while O<sub>3</sub> is VOCs-limited, indicating that NO<sub>x</sub> reductions would decrease SA, but VOCs-focused measures must be maintained to avoid O<sub>3</sub> rebounds. In June, both pollutants are VOCs-sensitive in most cities, making AVOCs reductions (industry + traffic) especially beneficial. In July, XA, XY, TC, and BJ occupy broader transitional zones, where reductions in NO<sub>x</sub> and AVOCs emissions both can yield benefits; WN's lingering VOC sensitivity suggests AVOC-focused strategies remain prioritized there. By August, SAFR in all cities tilting toward NO<sub>x</sub> sensitivity and O<sub>3</sub> is largely NO<sub>x</sub>-limited, suggesting that NO<sub>x</sub> reductions—particularly from power plants and traffic exhausts—become the most effective co-control approach.

From a policy perspective, warm-season air quality management in the GZB should adopt sub-seasonal sequencing emission controls. AVOCs reductions from traffic and industrial sources are prioritized in early summer, combined NO<sub>X</sub>–AVOCs strategies in transitional regimes are required in mid-summer, and NO<sub>X</sub> reductions to maximize co-benefits for both O<sub>3</sub> and SA are emphasized in late summer. Embedding SA considerations into an O<sub>3</sub>-focused framework allows decision-makers to capitalize on synergistic effects where they naturally arise, while avoiding unintended pollutant trade-offs, ultimately providing a more efficient pathway toward cleaner warm season air in the GZB.

## 4 Summary

This study combines long-term near-surface observations and high-resolution WRF-Chem simulations to diagnose summertime O<sub>3</sub> formation and its source attributions across the GZB, and to explore co-control potential with SA. Observational analyses show a nearly 50% decline in PM<sub>2.5</sub> mass concurrent with a marked rise in MDA8 O<sub>3</sub> concentrations and increasing O<sub>3</sub> exceedance frequency during warm seasons from 2014 to 2024, with June as the climatological O<sub>3</sub> peak month.

Incorporating an EKMA framework and BFM into WRF-Chem to simulate warm-season O<sub>3</sub> pollution in 2022, we find pronounced spatiotemporal shifts in OFR across the GZB: predominantly VOCs-limited in May–June, shifting toward transitional in July, and becoming NO<sub>x</sub>-limited in August. Sectoral attribution indicates contribution of anthropogenic sources to MDA8 O<sub>3</sub> concentrations rises from 32.7% to 55.2% (July), with biogenic shares peaking (18.7%) in July. City-level differences are notable—XA and XY are high anthropogenic cores with large industrial and traffic influences, WN shows the most and persistent VOCs-limited / NO<sub>x</sub>-saturated behavior, while BJ on the western edge is comparatively less polluted and

more NO<sub>X</sub>-limited or transitional. SAFR diagnostics shows a different seasonal pattern: the SAFR is NO<sub>X</sub>-limited in May, VOCs-limited in June, transitional in July, and transitional with a NO<sub>X</sub>-leaning sensitivity in August across the GZB. Traffic and industry emerge as primary contributors to both O<sub>3</sub> and SA, while power plants strongly influence SA in WN.

Based on these findings, we recommend sub-seasonally adaptive, city-specific control strategies: prioritize reductions of AVOCs, particularly from industrial and traffic sources, during early summer (notably June) to capture a VOCs-led co-benefit opportunity; implement combined NO<sub>X</sub>-AVOCs emission mitigation measures during the transitional month (July); and emphasize NO<sub>X</sub> mitigation from traffic, power plants emissions in late summer (August) to maximize co-benefits for both O<sub>3</sub> and SA while minimizing unintended trade-offs. Integrating real-time OFR and SAFR diagnostics with source attributions will enhance the efficiency and resilience of warm-season air quality management in the GZB.

Acknowledgments. This work is financially supported by the National Natural Science Foundation of China grant 42307154, the Key Research and Development Program of Shaanxi grant 2024SF-ZDCYL-05-05 and the China Postdoctoral Science Foundation grant 2023M743462.

Code and data availability. The hourly ambient surface O<sub>3</sub>, NO<sub>2</sub> and PM<sub>2.5</sub> mass concentrations are real-timely released by Ministry of Environmental Protection, China on the website http://www.aqistudy.cn/ (China MEP, 2013; last access: 20 August 2025). Observations of the near surface meteorological factors are released from <a href="http://www.meteomanz.com">http://www.meteomanz.com</a> (last access: 23 August 2025).

Author contributions. NZ, as the corresponding author, provided the ideas, verified the conclusions, and revised the paper. RW conducted research, designed the experiments, performed the simulation, processed the data, prepared the data visualization, and prepared the manuscript, with contributions from all authors. JW provided the primary data processing and reviewed the manuscript. JY, QJ, and YL analyzed the initial simulation data and visualized the model results. XT reviewed the manuscript and provided critical reviews.

**Competing interests.** The authors declare that they have no conflict of interest.

375

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

Table 1: Warm-season exceedance frequency and mean concentration of MDA8 O<sub>3</sub> in the GZB and its Cities from 2014 to 2024.

|      | Exceedance frequency |       |       |       |       |       | the Mean concentration (µg m <sup>-3</sup> ) |       |       |       |       |       |
|------|----------------------|-------|-------|-------|-------|-------|----------------------------------------------|-------|-------|-------|-------|-------|
|      | XA                   | XY    | WN    | TC    | BJ    | GZB   | XA                                           | XY    | WN    | TC    | BJ    | GZB   |
| 2014 | 9.8%                 | 8.5%  | 5.7%  | 12.6% | 5.0%  | 8.3%  | 175.3                                        | 186.7 | 182.2 | 183.2 | 178.0 | 179.0 |
| 2015 | 14.3%                | 8.1%  | 7.1%  | 12.2% | 6.1%  | 10.4% | 179.3                                        | 176.0 | 177.0 | 175.5 | 178.8 | 178.1 |
| 2016 | 22.4%                | 26.8% | 23.8% | 28.0% | 16.5% | 22.3% | 190                                          | 187.2 | 179.6 | 178.0 | 175.0 | 183.3 |
| 2017 | 43.1%                | 49.2% | 43.3% | 30.5% | 19.7% | 36.7% | 194.6                                        | 199.5 | 186.7 | 184.5 | 171.2 | 188.8 |
| 2018 | 31.3%                | 38.5% | 30.7% | 27.2% | 11.6% | 26.5% | 190.6                                        | 201.7 | 179.4 | 181.7 | 172.1 | 184.5 |
| 2019 | 25.2%                | 24.1% | 26.8% | 20.7% | 3.7%  | 19.4% | 183.6                                        | 178.1 | 176.9 | 174.7 | 171.3 | 178.1 |
| 2020 | 17.1%                | 17.9% | 13.4% | 13.6% | 6.3%  | 13.6% | 185.7                                        | 179.7 | 176.9 | 182.6 | 178.3 | 181.8 |
| 2021 | 20.9%                | 24.7% | 25.2% | 17.3% | 8.3%  | 18.2% | 194.8                                        | 188.5 | 187.5 | 179.7 | 173.2 | 185.9 |
| 2022 | 34.1%                | 41.5% | 22.8% | 16.8% | 15.9% | 26.7% | 185.5                                        | 181.7 | 179.7 | 176.5 | 174.9 | 180.6 |
| 2023 | 32.7%                | 34.1% | 17.9% | 16.5% | 13.4% | 23.7% | 187.2                                        | 185.7 | 183.4 | 177.7 | 174.2 | 182.0 |
| 2024 | 33.8%                | 42.3% | 27.9% | 20.7% | 14.0% | 27.0% | 180.6                                        | 184.7 | 176.4 | 174.1 | 174.5 | 178.2 |

Table 2: MDA8 O<sub>3</sub> contribution of various sources in urban areas of the GZB and five cities during warm season in 2022.

| Month  | Region | Anthro | Bio    | Ind    | Pow     | Tra    | Res   |
|--------|--------|--------|--------|--------|---------|--------|-------|
|        | GZB    | 32.77% | 9.74%  | 5.41%  | -4.39%  | 5.81%  | 3.30% |
|        | XA     | 37.14% | 10.46% | 6.43%  | -2.40%  | 5.02%  | 3.92% |
| M      | XY     | 35.69% | 10.73% | 4.64%  | -3.59%  | 4.77%  | 3.16% |
| May    | WN     | 13.95% | 9.05%  | 5.84%  | -21.82% | 7.59%  | 3.32% |
|        | TC     | 29.80% | 8.90%  | 1.22%  | 0.32%   | 7.07%  | 2.36% |
|        | BJ     | 29.32% | 7.13%  | 4.13%  | -0.89%  | 7.93%  | 1.99% |
|        | GZB    | 47.92% | 11.52% | 14.39% | 1.29%   | 13.61% | 4.07% |
|        | XA     | 52.71% | 11.71% | 17.44% | 3.56%   | 13.41% | 4.89% |
| June   | XY     | 52.67% | 11.23% | 14.31% | 3.99%   | 14.56% | 4.11% |
| June   | WN     | 27.44% | 11.85% | 10.90% | -18.17% | 13.42% | 3.16% |
|        | TC     | 42.58% | 11.92% | 6.27%  | 5.87%   | 13.48% | 2.97% |
|        | BJ     | 39.96% | 11.08% | 8.95%  | 1.60%   | 12.93% | 2.28% |
|        | GZB    | 55.23% | 18.67% | 11.80% | 3.67%   | 16.95% | 3.11% |
|        | XA     | 58.74% | 18.97% | 13.15% | 6.26%   | 16.76% | 2.80% |
| Luly   | XY     | 59.21% | 19.89% | 12.11% | 6.70%   | 17.04% | 4.18% |
| July   | WN     | 34.69% | 16.61% | 9.40%  | -18.13% | 15.49% | 2.61% |
|        | TC     | 51.43% | 19.44% | 5.86%  | 9.28%   | 15.91% | 2.30% |
|        | BJ     | 52.11% | 17.15% | 10.00% | 3.54%   | 18.34% | 2.89% |
|        | GZB    | 47.97% | 16.73% | 10.44% | 3.08%   | 15.74% | 2.57% |
|        | XA     | 51.09% | 16.45% | 11.01% | 4.41%   | 15.44% | 2.35% |
| Angust | XY     | 52.33% | 18.14% | 11.08% | 5.61%   | 16.21% | 2.31% |
| August | WN     | 34.63% | 18.46% | 10.62% | -9.05%  | 15.91% | 2.55% |
|        | TC     | 41.73% | 15.87% | 3.45%  | 7.37%   | 14.45% | 1.85% |
|        | BJ     | 43.35% | 14.48% | 9.04%  | 3.32%   | 16.00% | 3.67% |

Note: XA, XY, WN, BJ, TC and GZB represent the urban areas in Xi'an, Xianyang, Weinan, Baoji, Tongchuan and the Guanzhong Basin respectively. Anthro, Ind, Tra, Pow, Res, and Bio represent total anthropogenic, industry, tranffic, power plants, residential, and biogenic source, respectively.

Table 3: SA contribution of various sources in urban areas of the GZB and five cities during warm season in 2022.

| Month  | Region | Anthro | Bio    | Ind    | Pow    | Tra    | Res    |
|--------|--------|--------|--------|--------|--------|--------|--------|
| May    | GZB    | 78.79% | 3.16%  | 19.78% | 17.28% | 23.90% | 15.69% |
|        | XA     | 80.04% | 4.07%  | 21.12% | 16.09% | 23.90% | 16.59% |
|        | XY     | 80.19% | 3.38%  | 18.10% | 16.44% | 25.08% | 16.09% |
|        | WN     | 74.98% | 1.53%  | 15.65% | 25.56% | 17.46% | 10.33% |
|        | TC     | 74.60% | 0.11%  | 19.11% | 16.18% | 24.07% | 13.95% |
|        | BJ     | 76.82% | 1.97%  | 21.86% | 15.58% | 27.33% | 17.08% |
|        | GZB    | 74.93% | 4.30%  | 21.10% | 15.96% | 29.22% | 19.80% |
|        | XA     | 76.56% | 4.20%  | 24.97% | 13.84% | 31.05% | 20.93% |
| June   | XY     | 75.94% | 4.41%  | 19.22% | 14.43% | 32.86% | 21.43% |
|        | WN     | 75.10% | 3.42%  | 15.12% | 31.38% | 18.39% | 12.37% |
|        | TC     | 70.29% | 1.83%  | 16.73% | 14.00% | 26.89% | 19.06% |
|        | BJ     | 68.67% | 5.75%  | 17.67% | 11.30% | 28.02% | 20.65% |
|        | GZB    | 85.42% | 2.31%  | 25.34% | 26.65% | 31.75% | 14.21% |
|        | XA     | 85.41% | 2.67%  | 26.56% | 23.26% | 32.77% | 14.88% |
| Tl-    | XY     | 85.96% | 2.46%  | 23.41% | 27.42% | 34.85% | 15.72% |
| July   | WN     | 86.48% | 2.61%  | 24.49% | 40.33% | 21.71% | 7.07%  |
|        | TC     | 82.43% | -0.16% | 24.91% | 27.66% | 30.61% | 14.63% |
|        | BJ     | 84.15% | 1.15%  | 25.57% | 21.79% | 34.15% | 16.86% |
|        | GZB    | 69.71% | 6.58%  | 20.01% | 18.98% | 22.05% | 14.28% |
|        | XA     | 69.12% | 8.42%  | 21.03% | 15.25% | 22.88% | 15.52% |
| Amount | XY     | 70.00% | 8.50%  | 18.43% | 18.06% | 24.36% | 14.70% |
| August | WN     | 74.76% | 3.83%  | 21.09% | 33.74% | 16.44% | 9.71%  |
|        | TC     | 63.98% | 1.75%  | 15.38% | 15.63% | 20.15% | 11.55% |
|        | BJ     | 66.20% | 2.89%  | 19.13% | 14.28% | 23.14% | 15.98% |

Note: XA, XY, WN, BJ, TC and GZB represent the urban areas in Xi'an, Xianyang, Weinan, Baoji, Tongchuan and the Guanzhong Basin respectively. Anthro, Ind, Tra, Pow, Res, and Bio represent total anthropogenic, industry, tranffic, power plants, residential, and biogenic source, respectively.

Figure 1: Map showing (a) the location of simulation domain in China, (b) WRF-Chem model simulation domain with topography. In (c), the filled blue circles represent centers of cities with ambient monitoring sites and the size of circles denotes the number of ambient monitoring sites of cities. The white boundary outlines the Guanzhong Basin (GZB), the focus region of this study, comprising five cities: Xian (XA), Xianyang (XY), Weinan (WN), Tongchuan (TC), and Baoji (BJ).

Figure 2: Interannual variations of the warm-season (May–August) mean (a) PM<sub>2.5</sub> and (b) MDA8 O<sub>3</sub> concentrations in the GZB during 2014–2024, based on observations from 33 national monitoring sites. Box plots show the distribution (25th–75th percentiles, mean, whiskers, and outliers), and solid connecting lines indicate annual means, highlighting long-term trends.

Figure 3: MDA8 O<sub>3</sub> & SA isopleth profiles (μg m<sup>-3</sup>) in urban areas of the GZB in high-O<sub>3</sub> pollution episode during (a) & (e) May, (b) & (f) June, (c) & (g) July, (d) & (h) August 2022. The VOC-limited and NOx-limited regimes are separated by the red ridge lines. (d) and (e) indicate the formation regime in the GZB are NO<sub>X</sub>-limited with no red ridge lines displayed in the isopleth at current emission mitigation scenarios.

Figure 4: MDA8 O<sub>3</sub> isopleth profiles (μg m<sup>-3</sup>) in urban areas of (\*1) XA, (\*2) XY, (\*3) WN, (\*4) TC, and (\*5) BJ in high-O<sub>3</sub> pollution episode during (a\*) May, (b\*) June, (c\*) July, (d\*) August 2022. The VOCs-limited and NOx-limited regimes are separated by the red ridge lines. (d4) and (d5) indicate the formation regime are NOx-limited with no red ridge lines displayed in the isopleth at current emission mitigation scenarios.

Figure 5: The mean daytime  $(08:00 - 20:00 \text{ BJT}) \text{ NO}_X$ , HO<sub>2</sub> and HO concentrations from May to August 2022 in urban areas of the GZB.

Figure 6: The spatial distribution of total VOCs (includes BVOCs and AVOCs) to NO<sub>X</sub> emission ratios in (a) May, (b) June, (c) July and (d) August 2022.

Figure 7: The spatial distribution of the mean MDA8 O<sub>3</sub> contribution from (a) & (c) & (e) & (g) total anthropogenic, and (b) & (d) & (f) & (h) biogenic sources from May to August 2022.

Figure 8: Mean MDA8 O<sub>3</sub> contributions from various sources in (a) May, (b) June, (c) July, and (b) August 2022 in urban areas of the GZB and five cities.

Figure 9: SA isopleth profiles ( $\mu g \, m^{-3}$ ) in urban areas of (\*1) XA, (\*2) XY, (\*3) WN, (\*4) TC, and (\*5) BJ in high-O<sub>3</sub> pollution episode during (a\*) May, (b\*) June, (c\*) July, (d\*) August 2022. The VOCs-limited and NOx-limited regimes are separated by the red ridge lines. (a\*), (c4) and (d\*) indicate the formation regime are NOx-limited with no red ridge lines displayed in the isopleth at current emission mitigation scenarios.