# Peer review of "Sub-seasonal and Spatial Variations in Ozone Formation and Cocontrol Potential for Secondary Aerosols in the Guanzhong Basin, Central China"

_EGUsphere, 2025_

## Author Comment (AC1)

**Reply to Anonymous Referee #2**

We thank the reviewer very much for the careful reading of our manuscript and helpful comments. We have revised the manuscript following the suggestions, as described below.

This manuscript investigates the sub-seasonal and spatial variability of ozone and secondary aerosol formation regimes in the Guanzhong Basin from May to August 2022 based on WRF-Chem simulations and EKMA curves. The key finding is a sub-seasonal progression in $O_3$ sensitivity (VOC-limited --> transitional --> $NO_x$-limited) and policy implications for month- and city-specific controls. While the novelty of the approach is somewhat limited, the analysis of sub-seasonal regime shifts is timely and carries important policy relevance for regional air quality management. The manuscript is generally well written, and the methods are sound. I recommend publication after the following major and minor issues are addressed.

1. The policy implications of this study are largely based on simulations for a single year (2022). However, 2022 was one of the warmest years in China since 1961, characterized by intense heatwaves and drought, and emissions in 2022 may also differ from other years (e.g., 2019 or 2024) due to COVID-related impacts. Given the strong sensitivity of ozone formation to both meteorology and emissions, it is unclear whether the diagnosed regime shifts and associated policy implications are representative of other years. The authors are encouraged to discuss the robustness of their conclusions to interannual variability in meteorology and emissions. Where feasible, additional support using observations combined with indicator-based methods or box-model analyses could help verify the ozone formation regimes in 2022 and assess their consistency with other recent years.

**Response:** We sincerely thank the reviewer for raising this critical point regarding the potential influence of interannual variability in meteorology and emissions on our conclusions, which are primarily based on simulations for 2022. We agree that ensuring the robustness of the diagnosed ozone formation regime (OFR) shifts is essential for

deriving reliable policy implications.

To address this concern, utilizing the multi-year satellite data, we have calculated the monthly mean formaldehyde-to-$NO_2$ ratio (FNR), a widely used proxy for near-surface $O_3$ sensitivity, over the GZB for the warm seasons of 2021, 2022, and 2023. The FNR is derived from tropospheric column densities of HCHO and $NO_2$ retrieved by the OMI sensor. We have added data description in Section 2.2 and results and discussions in Section 3.3.1 as follows,

**L87-91:** "*To evaluate the robustness of the simulated OFR, we have employed satellite-derived column densities of formaldehyde (HCHO) and $NO_2$. Daily Level-3 gridded HCHO data are obtained from the OMI/Aura HCHO Total Column Daily L3 Global $0.1°×0.1°$ product (OMHCHOd v003; NASA GES DISC). Daily Level-3 gridded $NO_2$ data are sourced from the OMI/Aura $NO_2$ Cloud-Screened Total and Tropospheric Column L3 Global $0.25°×0.25°$ product (OMNO2d v003; NASA GES DISC). A grid cell is excluded from the monthly average calculation if valid data are available for fewer than 8 days in that month.*"

**L191-210:** "*To assess the robustness of the simulated sub-seasonal OFR progression against interannual variability in meteorology and emissions, we examined the formaldehyde-to-$NO_2$ ratio (FNR) from satellite retrievals for the GZB region over three consecutive warm seasons (2021-2023). FNR is a widely used indicator for inferring near-surface $O_3$ sensitivity, with thresholds typically defined as: FNR < 1 for VOCs-limited, 1–2 for transitional, and >2 for $NO_X$-limited regimes (Jin et al., 2015; Hata et al., 2025; Rahman et al., 2025). The monthly FNRs reveal a consistent sub-seasonal evolution pattern across the three years. The spatial distributions transition from being dominated by blue grids (low FNR, VOCs-limited) in early summer to green (transitional) and eventually yellow/red grids ($NO_X$-sensitive) by late summer, particularly evident in 2021 and 2022 (Fig. S6). At the basin scale, the mean FNR increased consistently from May to August, from 0.90 to 1.61 in 2021 and from 0.91 to 1.77 in 2022, reflecting a systematic seasonal shift toward more $NO_X$-limited $O_3$*

*formation (Table S3). Despite data gaps in May and August 2023, FNR values of 1.20 in June and 1.43 in July indicate a similar transition from transitional to more $NO_X$-sensitive conditions. Note that although column-based FNR is a useful indicator of surface $O_3$ sensitivity, satellite retrievals are subject to substantial uncertainties arising from measurement errors, cloud contamination, surface reflectivity, profile assumptions, and aerosol effects (Jin et al., 2017; Souri et al., 2023).*

*This independent, multi-year satellite evidence provides strong support for the central finding of our model-based analysis, namely a recurring sub-seasonal transition in $O_3$ formation regimes over the GZB, evolving from VOCs-limited conditions in early summer to transitional and ultimately $NO_X$-limited regimes by late summer. The consistency of this progression across years with contrasting meteorological conditions, including the extreme heat in 2022, indicates that the diagnosed regime shift is a robust characteristic of the regional photochemical environment. This behavior is therefore more plausibly attributable to recurrent seasonal drivers, such as increasing solar radiation, temperature, and biogenic emissions, rather than to anomalies associated with any single year."*

[Figure]

**Figure S6: Spatial distribution of the satellite-derived formaldehyde-to-NO$_2$ ratio (FNR) over the GZB and surrounding regions.** Monthly mean FNR values for the warm seasons (May–August) of (a–d) 2021, (e–h) 2022, and (i–l) 2023 are shown. FNR is calculated from tropospheric column densities of HCHO and NO$_2$ retrieved by the OMI sensor. According to typical threshold ranges applied in China, FNR values below 1.0 (blue tones) generally indicate VOCs-limited ozone formation regimes, values between 1.0 and 2.0 (green-yellow tones) indicate transitional regimes, and values above 2.0 (orange-red tones) indicate NO$_X$-limited regimes. Data gaps (white areas) are primarily due to cloud cover affecting the satellite retrievals.

**Table S3 : The monthly FNR derived from satellite retrievals averaged in the GZB during warm-seasons from 2021 to 2023.**

| FNR | May | June | July | August |
|---|---|---|---|---|
| 2021 | 0.90 | 1.19 | 1.20 | 1.61 |
| 2022 | 0.91 | 1.31 | 1.50 | 1.77 |
| 2023 | 0.97 | 1.20 | 1.43 | 1.51 |

**Reference:**

Hata, H., Inoue, K., Yoshikado, H., Genchi, Y., and Tsunemi, K.: Impact of introducing electric vehicles on ground-level O$_3$ and PM$_{2.5}$ in the Greater Tokyo Area: yearly trends and the importance of changes in the Urban Heat Island effect, Atmos. Chem. Phys., 25, 1037–1056, https://doi.org/10.5194/acp-25-1037-2025, 2025

Jin, X., and Holloway, T.: Spatial and temporal variability of ozone sensitivity over China observed from the Ozone Monitoring Instrument, J. Geophys. Res.-Atmos., 120, 7229–7246, https://doi.org/10.1002/2015JD023250, 2015.

Jin, X., Pusede, E. A., Wiedinmyer, C. J., Fischer, M. L., and Oetjen, H. J.: Evaluating a space-based indicator of surface ozone–NO$_x$–VOC sensitivity over midlatitude source regions and application to decadal trends, J. Geophys. Res.-Atmos., 122, 10451–10471, https://doi.org/10.1002/2017JD026720, 2017.

Rahman, M. M., Shults, R., and Ali, M. F.: Formaldehyde-to-nitrogen dioxide ratio (FNR) analysis for ozone sensitivity: a case study over Bangladesh using OMI data, Air Qual. Atmos. Health, 18, 1879–1886, https://doi.org/10.1007/s11869-025-01732-5, 2025.

Souri, A. H., Johnson, M. S., Wolfe, G. M., Crawford, J. H., Fried, A., Wisthaler, A., Brune, W. H., Blake, D. R., Weinheimer, A. J., Verhoelst, T., Compernolle, S., Pinardi, G., Vigouroux, C., Langerock, B., Choi, S., Lamsal, L., Zhu, L., Sun, S., Cohen, R. C., Min, K.-E., Cho, C., Philip, S., Liu, X., and Chance, K.: Characterization of errors in satellite-based HCHO/NO$_2$ tropospheric column ratios with respect to chemistry, column-to-PBL translation, spatial representation, and retrieval uncertainties, Atmos. Meas. Tech., 16, 1961–1986, https://doi.org/10.5194/amt-16-1961-2023, 2023.

2. The manuscript discusses the interactions between $PM_{2.5}$ and $O_3$, including aerosol radiative effects and heterogeneous uptake of $HO_2$ on $O_3$ production, but these interactions are not considered in the interpretation of ozone and secondary aerosol formation regimes. It would be helpful to discuss to what extent the substantial reduction in $PM_{2.5}$ mass from 2014–2024 may have contributed to changes in $O_3$ pollution, whether aerosol chemical and radiative effects affect the diagnosed sub-seasonal $O_3$ regimes, and how this might inform coordinated co-control strategies.

**Response:** We thank the reviewer for raising this important point regarding the potential role of aerosol–radiation–chemistry interactions in shaping $O_3$ trends and regimes, and their implications for coordinated control. We have added a new subsection (3.5.1 Impacts of Aerosol-Radiation-Chemistry Interactions) to thoroughly address these mechanisms.

**L338-367: "3.5.1 Impacts of Aerosol-Radiation-Chemistry Interactions**

*The substantial decline in $PM_{2.5}$ mass in the GZB over the past decade raises the question of whether aerosol–radiation and aerosol–chemistry interactions have notably influenced the observed increase in warm-season $O_3$ and possibly modulated its formation sensitivity. To quantify these effects, we conduct sensitivity experiments to separately isolate (i) aerosol-induced radiative changes (A_Rad) and (ii) changes in heterogeneous $HO_2\cdot$ uptake on wet aerosol surfaces (A_HO₂) associated with aerosol loading variations during the warm season from 2014 to 2022. In the A_Rad experiment, all model configurations are identical to the base simulation, except that aerosol concentrations within the PBL are fixed at their 2014 levels in the aerosol–radiation transfer module. Similarly, in the A_HO₂ experiment, aerosol concentrations are fixed at 2014 levels only in the calculation of heterogeneous $HO_2\cdot$ uptake on wet aerosol surfaces, while all other processes remained unchanged. The resulting differences from the base case therefore represent the impacts of aerosol changes between 2014 and 2022 on MDA8 $O_3$ through radiative and $HO_2\cdot$ heterogeneous loss pathways, respectively.*

*Over the period 2014–2022, during which observed $PM_{2.5}$ concentrations in the*

*GZB have declined by approximately 21.4 $\mu g\ m^{-3}$, the A_Rad and A_HO$_2$ effects exerted comparable influences on MDA8 O$_3$, each contributing between 3 and 7 $\mu g\ m^{-3}$ across most of the region. The combined influence of these two pathways results in an increase of approximately 4.0 $\mu g\ m^{-3}$ in the mean warm-season MDA8 O$_3$ concentration over the GZB, with peak enhancements exceeding 7 $\mu g\ m^{-3}$ in urban core areas with high aerosol levels during June and July (Fig. 9). Although non-negligible, this aerosol-mediated increase accounts for only 10.4 % of the total observed MDA8 O$_3$ rise (38.14 $\mu g\ m^{-3}$) during the warm seasons from 2014-2022 over the GZB. Thus, while the PM$_{2.5}$ cleanup has exerted a discernible upward pressure on O$_3$ via enhanced photochemistry and modified radical cycling, it is not the dominant driver of the worsening O$_3$ pollution; the primary factors remain the increased frequency of unfavorable synoptic conditions and changes in anthropogenic precursor emissions (Bei et al., 2022; Zhao et al., 2026).*

*A key follow-up question is whether these aerosol effects alter the identification of sub-seasonal OFRs. Additional sensitivity simulations for 2022, in which A_Rad and A_HO$_2$ are deliberately switched off, show that the combined A_Rad+A_HO$_2$ effect modestly suppresses MDA8 O$_3$ concentrations by 0.1-1.0 $\mu g\ m^{-3}$ under current emission levels but does not change the fundamental spatiotemporal progression of the O$_3$ formation sensitivity (Fig. S7). This indicates that the chemical sensitivity of O$_3$ production to its precursors is primarily governed by the evolving balance between NO$_X$ and VOCs under the prevailing meteorology, rather than by aerosol-mediated perturbations under contemporary pollution levels.*

*Therefore, although the historical PM$_{2.5}$ reduction has provided a modest boost to O$_3$ concentrations, it has not reshaped the intrinsic, meteorologically-driven transitions in OFR. This finding supports the use of OFR diagnostics, which are largely insensitive to aerosol loading in the current environment, as a reliable basis for designing seasonally adaptive control strategies. "*

[Figure]

**Figure 9: Impacts of aerosol changes on warm-season (May–August) MDA8 O₃ concentrations over the GZB during 2014–2022.** Panels (a), (d), (g), and (j) show the changes in MDA8 O₃ attributable to aerosol-radiation effects associated with aerosol variations for May, June, July, and August, respectively. Panels (b), (e), (h), and (k) show the corresponding MDA8 O₃ changes driven by variations in heterogeneous uptake of HO₂ radicals on wet aerosol surfaces induced by aerosol changes. Panels (c), (f), (i), and (l) present the combined effects of aerosol-radiation interactions and HO₂ heterogeneous uptake changes on MDA8 O₃ concentrations for the corresponding months.

[Figure]

**Figure S7:** MDA8 O₃ isopleth profiles (µg m⁻³) and corresponding MDA8 O₃ concentration variations in urban areas of the GZB during high-O₃ pollution episodes in 2022, with aerosol-radiation effects and heterogeneous uptake of HO₂ radicals on wet aerosol surfaces associated with aerosol variations excluded. Panels (a) and (e) show results for May, (b) and (f) for June, (c) and (g) for July, and (d) and (h) for August.

**Reference:**

Zhao, C., Sun, Y., Yang, J., Li, J., Zhou, Y., Yang, Y., Fan, H., and Zhao, X.: Decadal evolution of aerosol-mediated ozone responses in Eastern China under clean-air actions and carbon-neutrality policies, Atmos. Chem. Phys., 26, 1301–1318, https://doi.org/10.5194/acp-26-1301-2026, 2026.

Bei, N., Liang, J., Li, X., and Wang, R.: Worsening summertime ozone pollution in the Guanzhong Basin, China from 2014 to 2018: impacts of synoptic conditions and anthropogenic emissions, Atmos. Environ., 274, 118974, doi:10.1016/j.atmosenv.2022.118974, 2022.

3. Other minor comments:

**19: dominate should be dominant**

**Response:** corrected.

**216–222: These sentences are repeated.**

**Response:** We have removed the repeated statements.

**277–289: This paragraph reads more like background rather than results.**

**Response:** We have removed the paragraph in the results.

\# 300: Table 4 should be Table 3.

**Response:** corrected.

Section 3.5: The discussion of secondary aerosol formation regimes is largely descriptive. Additional explanation of the chemical or meteorological processes driving variability in SA regimes would improve the interpretation.

**Response:** We have added related discussion in Section 3.5 as follow,

**L379-388:** "*The observed spatiotemporal evolution of SAFR can be interpreted in the context of the seasonal progression of key chemical and meteorological drivers. The $NO_X$-limited regime in early summer is likely associated with an enhanced contribution of nitrate to secondary aerosols, under conditions where SA formation remains sensitive to $NO_X$ through $HNO_3$ production and subsequent gas–particle partitioning favored by relatively lower temperatures, higher humidity, and weaker photochemical activity. As solar radiation and temperature increase in June, the enhanced atmospheric oxidation capacity, together with temperature-dependent VOCs and biogenic emissions, promotes SOA formation, leading to a shift toward a VOCs-limited SAFR. The persistent VOCs sensitivity in WN into July reflects its relatively high $NO_X$ emissions, which decrease the local VOCs/$NO_X$ ratio and thereby reinforce VOCs-limited chemistry. By late summer (August), warmer and more humid conditions increase aerosol liquid water content and favor efficient nitrate formation and partitioning, contributing to a renewed tendency toward $NO_X$-sensitive SAFR.*"

Figure S4: The x-axis time label is incorrectly marked as 2018.

**Response:** We have revised the figures as follows,

[Figure]

**Figure S4:** Diurnal profiles of measured (black dots) and predicted (red line) (a) PM$_{2.5}$, (b) O$_3$, (c) NO$_2$, (d) SO$_2$, and (e) CO concentrations averaged over all ambient monitoring stations in the GZB from May to August 2022.

[Figure]

**Figure S2:** Temporal variations of predicted (red) and observed (black) (a) temperature at 2 m, (b) relative humid at 2 m, (c) wind speed and (d) wind direction at 10 m at Jinghe meteorological monitoring site from May to August 2022. The model performance statistic metrics of MB, RMSE and IOA are also shown.